# You Only Look at One Sequence: Rethinking Transformer in Vision through Object Detection

**Yuxin Fang** [1*]  **Bencheng Liao** [1*]  **Xinggang Wang** [1†]  **Jiemin Fang** [2,1]

**Jiyang Qi** [1]  **Rui Wu** [3]  **Jianwei Niu** [3]  **Wenyu Liu** [1]

[1] School of EIC, Huazhong University of Science & Technology
[2] Institute of AI, Huazhong University of Science & Technology
[3] Horizon Robotics
{yxf, bcliao, xgwang}@hust.edu.cn

## Abstract

Can Transformer perform 2D object- and region-level recognition from a pure sequence-to-sequence perspective with minimal knowledge about the 2D spatial structure? To answer this question, we present You Only Look at One Sequence (YOLOS), a series of object detection models based on the vanilla Vision Transformer with the fewest possible modifications, region priors, as well as inductive biases of the target task. We find that YOLOS pre-trained on the mid-sized ImageNet-$1k$ dataset *only* can already achieve quite competitive performance on the challenging COCO object detection benchmark, *e.g.*, YOLOS-Base directly adopted from BERT-Base architecture can obtain $42.0$ box AP on COCO `val`. We also discuss the impacts as well as limitations of current pre-train schemes and model scaling strategies for Transformer in vision through YOLOS. Code and pre-trained models are available at https://github.com/hustvl/YOLOS.

## 1 Introduction

Transformer [58] is born to transfer. In natural language processing (NLP), the dominant approach is to first pre-train Transformer on large, generic corpora for general language representation learning, and then fine-tune or adapt the model on specific target tasks [18]. Recently, Vision Transformer (ViT) [1] [21] demonstrates that canonical Transformer encoder architecture directly inherited from NLP can perform surprisingly well on image recognition at scale using modern vision transfer learning recipe [33]. Taking sequences of image patch embeddings as inputs, ViT can successfully transfer pre-trained general visual representations from sufficient scale to more specific image classification tasks with fewer data points from a pure sequence-to-sequence perspective.

Since a pre-trained Transformer can be successfully fine-tuned on sentence-level tasks [7, 19] in NLP, as well as *token-level* tasks [48, 52], where models are required to produce fine-grained output at the token-level [18]. A natural question is: Can ViT transfer to more challenging *object- and region-level* target tasks in computer vision such as object detection other than image-level recognition?

ViT-FRCNN [6] is the first to use a pre-trained ViT as the backbone for a Faster R-CNN [50] object detector. However, this design cannot get rid of the reliance on convolutional neural networks (CNNs)

---

*Yuxin Fang and Bencheng Liao contributed equally. †Xinggang Wang is the corresponding author. This work was done when Yuxin Fang was interning at Horizon Robotics mentored by Rui Wu.

[1] There are various sophisticated or hybrid architectures termed as "Vision Transformer". For disambiguation, in this paper, "Vision Transformer" and "ViT" refer to the canonical or vanilla Vision Transformer architecture proposed by Dosovitskiy et al. [21] unless specified.

and strong 2D inductive biases, as ViT-FRCNN re-interprets the output sequences of ViT to 2D spatial feature maps and depends on region-wise pooling operations (*i.e.,* RoIPool [23, 25] or RoIAlign [27]) as well as region-based CNN architectures [50] to decode ViT features for object- and region-level perception. Inspired by modern CNN design, some recent works [39, 59, 62, 65] introduce the pyramidal feature hierarchy, spatial locality, equivariant as well as invariant representations [24] to canonical Vision Transformer design, which largely boost the performance in dense prediction tasks including object detection. However, these architectures are performance-oriented and cannot reflect the properties of the canonical or vanilla Vision Transformer [21] directly inherited from Vaswani et al. [58]. Another series of work, the DEtection TRansformer (DETR) families [10, 72], use a random initialized Transformer to encode & decode CNN features for object detection, which does not reveal the transferability of a pre-trained Transformer.

Intuitively, ViT is designed to model long-range dependencies and global contextual information instead of local and region-level relations. Moreover, ViT lacks hierarchical architecture as modern CNNs [26, 35, 53] to handle the large variations in the scale of visual entities [1, 37]. Based on the available evidence, it is still unclear whether a pure ViT can transfer pre-trained general visual representations from image-level recognition to the much more complicated 2D object detection task.

To answer this question, we present You Only Look at One Sequence (YOLOS), a series of object detection models based on the canonical ViT architecture with the fewest possible modifications, region priors, as well as inductive biases of the target task injected. Essentially, the change from a pre-trained ViT to a YOLOS detector is embarrassingly simple: (1) YOLOS replaces one [CLS] token for image classification in ViT with one hundred [DET] tokens for object detection. (2) YOLOS replaces the image classification loss in ViT with the bipartite matching loss to perform object detection in a set prediction manner following Carion et al. [10], which can avoid re-interpreting the output sequences of ViT to 2D feature maps as well as prevent manually injecting heuristics and prior knowledge of object 2D spatial structure during label assignment [71]. Moreover, the prediction head of YOLOS can get rid of complex and diverse designs, which is as compact as a classification layer.

Directly inherited from ViT [21], YOLOS is not designed to be yet another high-performance object detector, but to unveil the versatility and transferability of pre-trained canonical Transformer from image recognition to the more challenging object detection task. Concretely, our main contributions are summarized as follows:

- We use the mid-sized ImageNet-1$k$ [51] as the *sole* pre-training dataset, and show that a vanilla ViT [21] can be successfully transferred to perform the complex object detection task and produce competitive results on COCO [36] benchmark with the fewest possible modifications, *i.e.*, by only looking at one sequence (YOLOS).

- For the first time, we demonstrate that 2D object detection can be accomplished in a pure sequence-to-sequence manner by taking a sequence of fixed-sized non-overlapping image patches as input. Among existing object detectors, YOLOS utilizes the minimal 2D inductive biases.

- For the vanilla ViT, we find the object detection results are quite sensitive to the pre-train scheme and the detection performance is far from saturating. Therefore the proposed YOLOS can be also used as a challenging benchmark task to evaluate different (label-supervised and self-supervised) pre-training strategies for ViT.

## 2   You Only Look at One Sequence

As for the model design, YOLOS closely follows the original ViT architecture [21], and is optimized for object detection in the same vein as Carion et al. [10]. YOLOS can be easily adapted to various canonical Transformer architectures available in NLP as well as in computer vision. This intentionally simple setup is not designed for better detection performance, but to exactly reveal characteristics of the Transformer family in object detection as unbiased as possible.

### 2.1   Architecture

An overview of the model is depicted in Fig. 1. Essentially, the change from a ViT to a YOLOS detector is simple: (1) YOLOS drops the [CLS] token for image classification and appends one

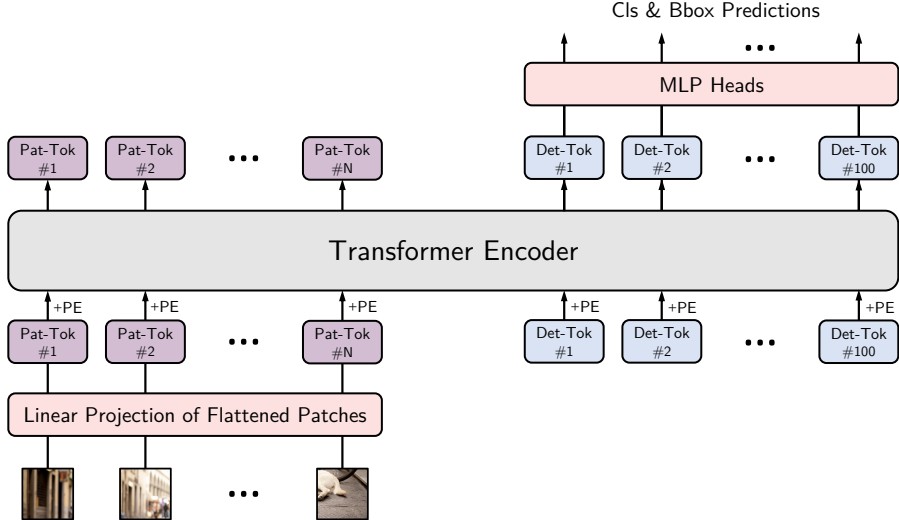

Figure 1: YOLOS architecture overview. "Pat-Tok" refers to [PATCH] token, which is the embedding of a flattened image patch. "Det-Tok" refers to [DET] token, which is a learnable embedding for object binding. "PE" refers to positional embedding. During training, YOLOS produces an optimal bipartite matching between predictions from one hundred [DET] tokens and ground truth objects. During inference, YOLOS directly outputs the final set of predictions in parallel. The figure style is inspired by Dosovitskiy et al. [21].

hundred randomly initialized learnable detection tokens ([DET] tokens) to the input patch embeddings ([PATCH] tokens) for object detection. (2) During training, YOLOS replaces the image classification loss in ViT with the bipartite matching loss to perform object detection in a set prediction manner following Carion et al. [10].

**Stem.** The canonical ViT [21] receives an 1D sequence of embedded tokens as the input. To handle 2D image inputs, we reshape the image $\mathbf{x} \in \mathbb{R}^{H \times W \times C}$ into a sequence of flattened 2D image patches $\mathbf{x}_{\text{PATCH}} \in \mathbb{R}^{N \times (P^2 \cdot C)}$. Here, $(H, W)$ is the resolution of the input image, $C$ is the number of input channels, $(P, P)$ is the resolution of each image patch, and $N = \frac{HW}{P^2}$ is the resulting number of patches. Then we map $\mathbf{x}_{\text{PATCH}}$ to $D$ dimensions with a trainable linear projection $\mathbf{E} \in \mathbb{R}^{(P^2 \cdot C) \times D}$. We refer to the output of this projection $\mathbf{x}_{\text{PATCH}}\mathbf{E}$ as [PATCH] tokens. Meanwhile, one hundred randomly initialized learnable [DET] tokens $\mathbf{x}_{\text{DET}} \in \mathbb{R}^{100 \times D}$ are appended to the [PATCH] tokens. Position embeddings $\mathbf{P} \in \mathbb{R}^{(N+100) \times D}$ are added to all the input tokens to retain positional information. We use the standard learnable 1D position embeddings following Dosovitskiy et al. [21]. The resulting sequence $\mathbf{z}_0$ serves as the input of YOLOS Transformer encoder. Formally:

$$\mathbf{z}_0 = \left[ \mathbf{x}_{\text{PATCH}}^1 \mathbf{E}; \cdots; \mathbf{x}_{\text{PATCH}}^N \mathbf{E}; \ \mathbf{x}_{\text{DET}}^1; \cdots; \mathbf{x}_{\text{DET}}^{100} \right] + \mathbf{P}. \tag{1}$$

**Body.** The body of YOLOS is basically the same as ViT, which consists of a stack of Transformer encoder layers only [58]. [PATCH] tokens and [DET] tokens are treated equally and they perform global interactions inside Transformer encoder layers.

Each Transformer encoder layer consists of one multi-head self-attention (MSA) block and one MLP block. LayerNorm (LN) [2] is applied before every block, and residual connections [26] are applied after every block [3, 61]. The MLP contains one hidden layer with an intermediate GELU [29] non-linearity activation function. Formally, for the $\ell$-*th* YOLOS Transformer encoder layer:

$$\begin{aligned} \mathbf{z}'_\ell &= \text{MSA}\left(\text{LN}\left(\mathbf{z}_{\ell-1}\right)\right) + \mathbf{z}_{\ell-1}, \\ \mathbf{z}_\ell &= \text{MLP}\left(\text{LN}\left(\mathbf{z}'_\ell\right)\right) + \mathbf{z}'_\ell. \end{aligned} \tag{2}$$

**Detector Heads.** The detector head of YOLOS gets rid of complex and heavy designs, and is as neat as the image classification layer of ViT. Both the classification and the bounding box regression heads are implemented by one MLP with separate parameters containing two hidden layers with intermediate ReLU [41] non-linearity activation functions.

**Detection Token.** We purposefully choose randomly initialized [DET] tokens as proxies for object representations to avoid inductive biases of 2D structure and prior knowledge about the task injected during label assignment. When fine-tuning on COCO, for each forward pass, an optimal bipartite matching between predictions generated by [DET] tokens and ground truth objects is established. This procedure plays the same role as label assignment [10, 71], but is unaware of the input 2D structure, *i.e.*, YOLOS does not need to re-interpret the output sequence of ViT to an 2D feature maps for label assignment. Theoretically, it is feasible for YOLOS to perform any dimensional object detection without knowing the exact spatial structure and geometry, as long as the input is always flattened to a sequence in the same way for each pass.

**Fine-tuning at Higher Resolution.** When fine-tuning on COCO, all the parameters are initialized from ImageNet-$1k$ pre-trained weights except for the MLP heads for classification & bounding box regression as well as one hundred [DET] tokens, which are randomly initialized. During fine-tuning, the image has a much higher resolution than pre-training. We keep the patch size $P$ unchanged, *i.e.*, $P \times P = 16 \times 16$, which results in a larger effective sequence length. While ViT can handle arbitrary input sequence lengths, the positional embeddings need to adapt to the longer input sequences with various lengths. We perform 2D interpolation of the pre-trained position embeddings on the fly[2].

**Inductive Bias.** We carefully design the YOLOS architecture for the minimal additional inductive biases injection. The inductive biases inherent from ViT come from the patch extraction at the network stem part as well as the resolution adjustment for position embeddings [21]. Apart from that, YOLOS adds no non-degenerated (*e.g.*, $3 \times 3$ or other non $1 \times 1$) convolutions upon ViT [3]. From the representation learning perspective, we choose to use [DET] tokens to bind objects for final predictions to avoid additional 2D inductive biases as well as task-specific heuristics. The performance-oriented design inspired by modern CNN architectures such as pyramidal feature hierarchy, 2D local spatial attention as well as the region-wise pooling operation is not applied. All these efforts are meant to exactly unveil the versatility and transferability of pre-trained Transformers from image recognition to object detection in a pure sequence-to-sequence manner, with minimal knowledge about the input spatial structure and geometry.

**Comparisons with DETR.** The design of YOLOS is deeply inspired by DETR [10]: YOLOS uses [DET] tokens following DETR as proxies for object representations to avoid inductive biases about 2D structures and prior knowledge about the task injected during label assignment, and YOLOS is optimized similarly as DETR.

Meanwhile, there are some key differences between the two models: (1) DETR adopts a Transformer encoder-decoder architecture, while YOLOS chooses an encoder-only Transformer architecture. (2) DETR only employs pre-training on its CNN backbone but leaves the Transformer encoder & decoder being trained from random initialization, while YOLOS naturally inherits representations from any pre-trained canonical ViT. (3) DETR applies cross-attention between encoded image features and object queries with auxiliary decoding losses deeply supervised at each decoder layer, while YOLOS always looks at only one sequence for each encoder layer, without distinguishing [PATCH] tokens and [DET] tokens in terms of operations. Quantitative comparisons between the two are in Sec. 3.4.

## 3 Experiments

### 3.1 Setup

**Pre-training.** We pre-train all YOLOS / ViT models on ImageNet-$1k$ [51] dataset using the data-efficient training strategy suggested by Touvron et al. [57]. The parameters are initialized with a truncated normal distribution and optimized using AdamW [40]. The learning rate and batch size are $1 \times 10^{-3}$ and 1024, respectively. The learning rate decay is cosine and the weight decay is 0.05. Rand-Augment [14] and random erasing [69] implemented by `timm` library [64] are used for data augmentation. Stochastic depth [32], Mixup [68] and Cutmix [66] are used for regularization.

---

[2]The configurations of position embeddings are detailed in the Appendix.

[3]We argue that it is imprecise to say Transformer do not have convolutions. All linear projection layers in Transformer are equivalent to point-wise or $1 \times 1$ convolutions with sparse connectivity, parameter sharing, and equivalent representations properties, which can largely improve the computational efficiency compared with the "all-to-all" interactions in fully-connected design that has even weaker inductive biases [5, 24].

**Fine-tuning.** We fine-tune all YOLOS models on COCO object detection benchmark [36] in a similar way as Carion et al. [10]. All the parameters are initialized from ImageNet-$1k$ pre-trained weights except for the MLP heads for classification & bounding box regression as well as one hundred [DET] tokens, which are randomly initialized. We train YOLOS on a single node with $8 \times 12\text{G}$ GPUs. The learning rate and batch sizes are $2.5 \times 10^{-5}$ and 8 respectively. The learning rate decay is cosine and the weight decay is $1 \times 10^{-4}$.

As for data augmentation, we use multi-scale augmentation, resizing the input images such that the shortest side is at least 256 and at most 608 pixels while the longest at most 864 for tiny models. For small and base models, we resize the input images such that the shortest side is at least 480 and at most 800 pixels while the longest at most 1333. We also apply random crop augmentations during training following Carion et al. [10]. The number of [DET] tokens are 100 and we keep the loss function as well as loss weights the same as DETR, while we don't apply dropout [54] or stochastic depth during fine-tuning since we find these regularization methods hurt performance.

**Model Variants.** With available computational resources, we study several YOLOS variants. Detailed configurations are summarized in Tab. 1. The input patch size for all models is $16 \times 16$. YOLOS-Ti (Tiny), -S (Small), and -B (Base) directly correspond to DeiT-Ti, -S, and -B [57]. From the model scaling perspective [20, 56, 60], the small and base models of YOLOS / DeiT can be seen as performing width scaling ($w$) [30, 67] on the corresponding tiny model.

| Model | DeiT [57] Model | Layers (Depth) | Embed. Dim. (Width) | Pre-train Resolution | Heads | Params. | FLOPs | $\frac{f(\texttt{Lin.})}{f(\texttt{Att.})}$ |
|---|---|---|---|---|---|---|---|---|
| YOLOS-Ti | DeiT-Ti | | 192 | | 3 | 5.7 M | 1.2 G | 5.9 |
| YOLOS-S | DeiT-S | 12 | 384 | 224 | 6 | 22.1 M | 4.5 G | 11.8 |
| YOLOS-B | DeiT-B | | 768 | | 12 | 86.4 M | 17.6 G | 23.5 |
| YOLOS-S ($dwr$) | – | 19 | 240 | 272 | 6 | 13.7 M | 4.6 G | 5.0 |
| YOLOS-S ($d\mathbf{w}r$) | – | 14 | 330 | 240 | 6 | 19.0 M | 4.6 G | 8.8 |

Table 1: Variants of YOLOS. "$dwr$" and "$d\mathbf{w}r$" refer to uniform compound model scaling and fast model scaling, respectively. The "$dwr$" and "$d\mathbf{w}r$" notations are inspired by Dollár et al. [20]. Note that all the numbers listed are for pre-training, which could change during fine-tuning, *e.g.*, the resolution and FLOPs.

Besides, we investigate two other model scaling strategies which proved to be effective in CNNs. The first one is uniform compound scaling ($dwr$) [20, 56]. In this case, the scaling is uniform w.r.t. FLOPs along all model dimensions (*i.e.*, width ($w$), depth ($d$) and resolution ($r$)). The second one is fast scaling ($d\mathbf{w}r$) [20] that encourages primarily scaling model width ($\mathbf{w}$), while scaling depth ($d$) and resolution ($r$) to a lesser extent w.r.t. FLOPs. During the ImageNet-$1k$ pre-training phase, we apply $dwr$ and $d\mathbf{w}r$ scaling to DeiT-Ti ($\sim 1.2\text{G}$ FLOPs) and scale the model to $\sim 4.5\text{G}$ FLOPs to align with the computations of DeiT-S. Larger models are left for future work.

For canonical CNN architectures, the model complexity or FLOPs ($f$) are proportional to $dw^2r^2$ [20]. Formally, $f(\texttt{CNN}) \propto dw^2r^2$. Different from CNN, there are two kinds of operations that contribute to the FLOPs of ViT. The first one is the linear projection (`Lin.`) or point-wise convolution, which fuses the information across different channels point-wisely via learnable parameters. The complexity is $f(\texttt{Lin.}) \propto dw^2r^2$, which is the same as $f(\texttt{CNN})$. The second one is the spatial attention (`Att.`), which aggregates the spatial information depth-wisely via computed attention weights. The complexity is $f(\texttt{Att.}) \propto dwr^4$, which grows quadratically with the input sequence length or number of pixels.

Note that the available scaling strategies are designed for architectures with complexity $f \propto dw^2r^2$, so theoretically the $dwr$ as well as $d\mathbf{w}r$ model scaling are not directly applicable to ViT. However, during pre-training phase the resolution is relatively low, therefore $f(\texttt{Lin.})$ dominates the FLOPs ($\frac{f(\texttt{Lin.})}{f(\texttt{Att.})} > 5$). Our experiments indicate that some model scaling properties of ViT are consistent with CNNs when $\frac{f(\texttt{Lin.})}{f(\texttt{Att.})}$ is large.

### 3.2 The Effects of Pre-training

We study the effects of different pre-training strategies (both label-supervised and self-supervised) when transferring ViT (DeiT-Ti and DeiT-S) from ImageNet-$1k$ to the COCO object detection benchmark via YOLOS. For object detection, the input shorter size is 512 for tiny models and is 800 for small models during inference. The results are shown in Tab. 2 and Tab. 3.

| Model | Pre-train Method | Pre-train Epochs | Fine-tune Epochs | Pre-train pFLOPs | Fine-tune pFLOPs | Total pFLOPs | ImNet Top-1 | AP |
|---|---|---|---|---|---|---|---|---|
| YOLOS-Ti | Rand. Init. | 0 | 600 | 0 | $14.2 \times 10^2$ | $14.2 \times 10^2$ | – | 19.7 |
|  | Label Sup. [57] | 200 |  | $3.1 \times 10^2$ |  | $10.2 \times 10^2$ | 71.2 | 26.9 |
|  | Label Sup. [57] | 300 | 300 | $4.7 \times 10^2$ | $7.1 \times 10^2$ | $11.8 \times 10^2$ | 72.2 | 28.7 |
|  | Label Sup. (🎓) [57] | 300 |  | $4.7 \times 10^2$ |  | $11.8 \times 10^2$ | 74.5 | 29.7 |
| YOLOS-S | Rand. Init. | 0 | 250 | 0 | $5.9 \times 10^3$ | $5.9 \times 10^3$ | – | 20.9 |
|  | Label Sup. [57] | 100 |  | $0.6 \times 10^3$ |  | $4.1 \times 10^3$ | 74.5 | 32.0 |
|  | Label Sup. [57] | 200 | 150 | $1.2 \times 10^3$ | $3.5 \times 10^3$ | $4.7 \times 10^3$ | 78.5 | 36.1 |
|  | Label Sup. [57] | 300 |  | $1.8 \times 10^3$ |  | $5.3 \times 10^3$ | 79.9 | 36.1 |
|  | Label Sup. (🎓) [57] | 300 |  | $1.8 \times 10^3$ |  | $5.3 \times 10^3$ | 81.2 | 37.2 |

Table 2: The effects of label-supervised pre-training. "pFLOPs" refers to petaFLOPs ($\times 10^{15}$). "ImNet" refers to ImageNet-$1k$. "🎓" refers to the distillation method from Touvron et al. [57].

| Model | Self Sup. Pre-train Method | Pre-train Epochs | Fine-tune Epochs | Linear Acc. | AP |
|---|---|---|---|---|---|
| YOLOS-S | MoCo-v3 [13] | 300 | 150 | 73.2 | 33.6 |
|  | DINO [11] | 800 | 150 | 77.0 | 36.2 |

Table 3: Study of self-supervised pre-training on YOLOS-S.

**Necessity of Pre-training.** At least under prevalent transfer learning paradigms [10, 57], the pre-training is necessary in terms of computational efficiency. For both tiny and small models, we find that pre-training on ImageNet-$1k$ saves the total theoretical forward pass computations (total pre-training FLOPs & total fine-tuning FLOPs) compared with training on COCO from random initialization (training from scratch [28]). Models trained from scratch with hundreds of epochs still lag far behind the pre-trained ViT even if given more total FLOPs budgets. This seems quite different from canonical modern CNN-based detectors, which can catch up with pre-trained counterparts quickly [28].

**Label-supervised Pre-training.** For supervised pre-training with ImageNet-$1k$ ground truth labels, we find that different-sized models prefer different pre-training schedules: 200 epochs pre-training for YOLOS-Ti still cannot catch up with 300 epochs pre-training even with a 300 epochs fine-tuning schedule, while for the small model 200 epochs pre-training provides feature representations as good as 300 epochs pre-training for transferring to the COCO object detection benchmark.

With additional transformer-specific distillation ("🎓") introduced by Touvron et al. [57], the detection performance is further improved by $\sim 1$ AP for both tiny and small models, in part because exploiting a CNN teacher [47] during pre-training helps ViT adapt to COCO better. It is also promising to directly leverage [DET] tokens to help smaller YOLOS learn from larger YOLOS on COCO during fine-tuning in a similar way as Touvron et al. [57], we leave it for future work.

**Self-supervised Pre-training.** The success of Transformer in NLP greatly benefits from large-scale self-supervised pre-training [18, 44, 45]. In vision, pioneering works [12, 21] train self-supervised Transformers following the masked auto-encoding paradigm in NLP. Recent works [11, 13] based on siamese networks show intriguing properties as well as excellent transferability to downstream tasks. Here we perform a preliminary transfer learning experiment on YOLOS-S using MoCo-v3 [13] and DINO [11] self-supervised pre-trained ViT weights in Tab. 3.

The transfer learning performance of $800$ epochs DINO self-supervised model on COCO object detection is on a par with 300 epochs DeiT label-supervised pre-training, suggesting great potentials of self-supervised pre-training for ViT on challenging object-level recognition tasks. Meanwhile, the transfer learning performance of MoCo-v3 is less satisfactory, in part for the MoCo-v3 weight is heavily under pre-trained. Note that the pre-training epochs of MoCo-v3 are the same as DeiT (300 epochs), which means that there is still a gap between the current state-of-the-art self-supervised pre-training approach and the prevalent label-supervised pre-training approach for YOLOS.

**YOLOS as a Transfer Learning Benchmark for ViT.** From the above analysis, we conclude that the ImageNet-$1k$ pre-training results cannot precisely reflect the transfer learning performance on COCO object detection. Compared with widely used image recognition transfer learning benchmarks such as CIFAR-10/100 [34], Oxford-IIIT Pets [43] and Oxford Flowers-102 [42], the performance of

YOLOS on COCO is more sensitive to the pre-train scheme and the performance is far from saturating. Therefore it is reasonable to consider YOLOS as a challenging transfer learning benchmark to evaluate different (label-supervised or self-supervised) pre-training strategies for ViT.

### 3.3 Pre-training and Transfer Learning Performance of Different Scaled Models

We study the pre-training and the transfer learning performance of different model scaling strategies, *i.e.*, width scaling ($w$), uniform compound scaling ($dwr$) and fast scaling ($d\mathbf{w}r$). The models are scaled from $\sim 1.2$G to $\sim 4.5$G FLOPs regime for pre-training. Detailed model configurations and descriptions are given in Sec. 3.1 and Tab. 1.

We pre-train all the models for 300 epochs on ImageNet-$1k$ with input resolution determined by the corresponding scaling strategies, and then fine-tune these models on COCO for 150 epochs. Few literatures are available for resolution scaling in object detection, where the inputs are usually oblong in shape and the multi-scale augmentation [10, 27] is used as a common practice. Therefore for each model during inference, we select the smallest resolution (*i.e.*, the shorter size) ranging in $[480, 800]$ producing the highest box AP, which is 784 for $dwr$ scaling and 800 for all the others. The results are summarized in Tab. 4.

| Scale | Image Classification @ ImageNet-$1k$ | | | | Object Detection @ COCO `val` | | | |
|---|---|---|---|---|---|---|---|---|
| | FLOPs | $\frac{f(\texttt{Lin.})}{f(\texttt{Att.})}$ | FPS | Top-1 | FLOPs | $\frac{f(\texttt{Lin.})}{f(\texttt{Att.})}$ | FPS | AP |
| $-$ | 1.2 G | 5.9 | 1315 | 72.2 | 81 G | 0.28 | 12.0 | 29.6 |
| $w$ | 4.5 G | 11.8 | 615 | 79.9 | 194 G | 0.55 | 5.7 | 36.1 |
| $dwr$ | 4.6 G | 5.0 | 386 | 80.5 | 163 G | 0.35 | 4.5 | 36.2 |
| $d\mathbf{w}r$ | 4.6 G | 8.8 | 511 | 80.4 | 172 G | 0.49 | 5.7 | 37.6 |

Table 4: Pre-training and transfer learning performance of different scaled models. FLOPs and FPS data of object detection are measured over the first 100 images of COCO `val` split during inference following Carion et al. [10]. FPS is measured with batch size 1 on a single 1080Ti GPU.

**Pre-training.** Both $dwr$ and $d\mathbf{w}r$ scaling can improve the accuracy compared with simple $w$ scaling, *i.e.*, the DeiT-S baseline. Other properties of each scaling strategy are also consistent with CNNs [20, 56], *e.g.*, $w$ scaling is the most speed friendly. $dwr$ scaling achieves the strongest accuracy. $d\mathbf{w}r$ is nearly as fast as $w$ scaling and is on a par with $dwr$ scaling in accuracy. Perhaps the reason why these CNN model scaling strategies are still appliable to ViT is that during pre-training the linear projection ($1 \times 1$ convolution) dominates the model computations.

**Transfer Learning.** The picture changes when transferred to COCO. The input resolution $r$ is much higher so the spatial attention takes over and linear projection part is no longer dominant in terms of FLOPs ($\frac{f(\texttt{Lin.})}{f(\texttt{Att.})} \propto \frac{w}{r^2}$). Canonical CNN model scaling recipes do not take spatial attention computations into account. Therefore there is some inconsistency between pre-training and transfer learning performance: Despite being strong on ImageNet-$1k$, the $dwr$ scaling achieves similar box AP as simple $w$ scaling. Meanwhile, the performance gain from $d\mathbf{w}r$ scaling on COCO cannot be clearly explained by the corresponding CNN scaling methodology that does not take $f(\texttt{Att.}) \propto dwr^4$ into account. The performance inconsistency between pre-training and transfer learning calls for novel model scaling strategies for ViT considering spatial attention complexity.

### 3.4 Comparisons with CNN-based Object Detectors

In previous sections, we treat YOLOS as a touchstone for the transferability of ViT. In this section, we consider YOLOS as an object detector and we compare YOLOS with some modern CNN detectors.

**Comparisons with Tiny-sized CNN Detectors.** As shown in Tab. 5, the tiny-sized YOLOS model achieves impressive performance compared with well-established and highly-optimized CNN object detectors. YOLOS-Ti is strong in AP and competitive in FLOPs & FPS even though Transformer is not intentionally designed to optimize these factors. From the model scaling perspective [20, 56, 60], YOLOS-Ti can serve as a promising model scaling start point.

**Comparisons with DETR.** The relations and differences in model design between YOLOS and DETR are given in Sec. 2.1, here we make quantitative comparisons between the two.

| Method | Backbone | Size | AP | Params. (M) | FLOPs (G) | FPS |
|---|---|---|---|---|---|---|
| YOLOv3-Tiny [49] | DarkNet [49] | $416 \times 416$ | 16.6 | 8.9 | 5.6 | 330 |
| YOLOv4-Tiny [60] | COSA [60] | $416 \times 416$ | 21.7 | 6.1 | 7.0 | 371 |
| **YOLOS-Ti** | DeiT-Ti (🐝) [57] | $256 \times *$ | 23.1 | 6.5 | 3.4 | 114 |
| CenterNet [70] | ResNet-18 [26] | $512 \times 512$ | 28.1 | – | – | 129 |
| YOLOv4-Tiny ($3l$) [60] | COSA [60] | $320 \times 320$ | 28.7 | – | – | 252 |
| Def. DETR [72] | FBNet-V3 [15] | $800 \times *$ | 27.9 | 12.2 | 12.3 | 35 |
| **YOLOS-Ti** | DeiT-Ti (🐝) [57] | $432 \times *$ | 28.6 | 6.5 | 11.7 | 84 |

Table 5: Comparisons with some tiny-sized modern CNN detectors. All models are trained to be fully converged. "Size" refers to input resolution for inference. FLOPs and FPS data are measured over the first 100 images of COCO `val` split during inference following Carion et al. [10]. FPS is measured with batch size 1 on a single 1080Ti GPU.

| Method | Backbone | Epochs | Size | AP | Params. (M) | FLOPs (G) | FPS |
|---|---|---|---|---|---|---|---|
| Def. DETR [72] | FBNet-V3 [15] | 150 | $800 \times *$ | 27.5 | 12.2 | 12.3 | 35 |
| **YOLOS-Ti** | DeiT-Ti [57] | 300 | $512 \times *$ | 28.7 | 6.5 | 18.8 | 60 |
| **YOLOS-Ti** | DeiT-Ti (🐝) [57] | 300 | $432 \times *$ | 28.6 | 6.5 | 11.7 | 84 |
| **YOLOS-Ti** | DeiT-Ti (🐝) [57] | 300 | $528 \times *$ | 30.0 | 6.5 | 20.7 | 51 |
| DETR [10] | ResNet-18-DC5 [26] | | $800 \times *$ | 36.9 | 29 | 129 | 7.4 |
| **YOLOS-S** | DeiT-S [57] | | $800 \times *$ | 36.1 | 31 | 194 | 5.7 |
| **YOLOS-S** | DeiT-S (🐝) [57] | 150 | $800 \times *$ | 37.2 | 31 | 194 | 5.7 |
| **YOLOS-S** ($d\mathbf{w}r$) | DeiT-S [57] ($d\mathbf{w}r$ Scale [20]) | | $704 \times *$ | 37.2 | 28 | 123 | 7.7 |
| **YOLOS-S** ($d\mathbf{w}r$) | DeiT-S [57] ($d\mathbf{w}r$ Scale [20]) | | $784 \times *$ | 37.6 | 28 | 172 | 5.7 |
| DETR [10] | ResNet-101-DC5 [26] | 150 | $800 \times *$ | 42.5 | 60 | 253 | 5.3 |
| **YOLOS-B** | DeiT-B (🐝) [57] | | | 42.0 | 127 | 538 | 2.7 |

Table 6: Comparisons with different DETR models. Tiny-sized models are trained to be fully converged. "Size" refers to input resolution for inference. FLOPs and FPS data are measured over the first 100 images of COCO `val` split during inference following Carion et al. [10]. FPS is measured with batch size 1 on a single 1080Ti GPU. The "ResNet-18-DC5" implantation is from `timm` library [64].

As shown in Tab. 6, YOLOS-Ti still performs better than the DETR counterpart, while larger YOLOS models with width scaling become less competitive: YOLOS-S with more computations is 0.8 AP lower compared with a similar-sized DETR model. Even worse, YOLOS-B cannot beat DETR with over $2\times$ parameters and FLOPs. Even though YOLOS-S with $d\mathbf{w}r$ scaling is able to perform better than the DETR counterpart, the performance gain cannot be clearly explained as discussed in Sec. 3.3.

**Interpreting the Results.** Although the performance is seemingly discouraging, the numbers are meaningful, as YOLOS is not purposefully designed for better performance, but designed to precisely reveal the transferability of ViT in object detection. *E.g.*, YOLOS-B is directly adopted from the BERT-Base architecture [18] in NLP. This 12 layers, 768 channels Transformer along with its variants have shown impressive performance on a wide range of NLP tasks. We demonstrate that with minimal modifications, this kind of architecture can also be successfully transferred (*i.e.*, AP $= 42.0$) to the challenging COCO object detection benchmark in computer vision from a pure sequence-to-sequence perspective. The minimal modifications from YOLOS exactly reveal the versatility and generality of Transformer.

### 3.5 Inspecting Detection Tokens

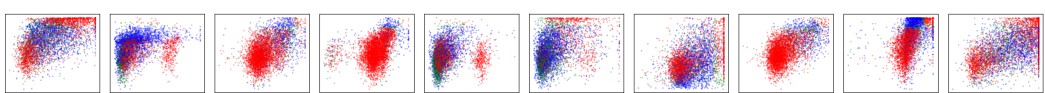

Figure 2: Visualization of all box predictions on all images from COCO `val` split for the first ten [DET] tokens. Each box prediction is represented as a point with the coordinates of its center normalized by each thumbnail image size. The points are color-coded so that **blue** points corresponds to small objects, **green** to medium objects and **red** to large objects. We observe that each [DET] token learns to specialize on certain regions and sizes. The visualization style is inspired by Carion et al. [10].

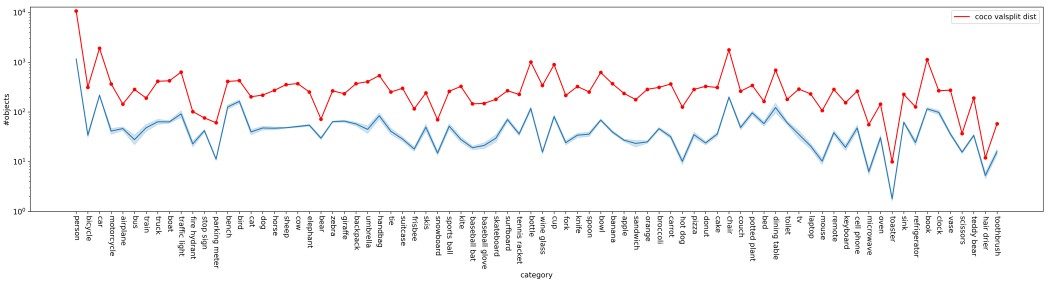

Figure 3: The statistics of all ground truth object categories (the **red** curve) and the statistics of all object category predictions from all $\boxed{\text{DET}}$ tokens (the **blue** curve) on all images from COCO `val` split. The error bar of the **blue** curve represents the variability of the preference of different tokens for a given category, which is small. This suggests that different $\boxed{\text{DET}}$ tokens are category insensitive.

**Qualitative Analysis on Detection Tokens.** As an object detector, YOLOS uses $\boxed{\text{DET}}$ tokens to represent detected objects. In general, we find that $\boxed{\text{DET}}$ tokens are sensitive to object locations and sizes, while insensitive to object categories, as shown in Fig. 2 and Fig. 3.

**Quantitative Analysis on Detection Tokens.** We give a quantitative analysis on the relation between $X = $ the cosine similarity of $\boxed{\text{DET}}$ token pairs, and $Y = $ the corresponding predicted bounding box centers $\ell_2$ distances. We use the Pearson correlation coefficient $\rho_{X,Y} = \frac{\mathbb{E}[(X-\mu_X)(Y-\mu_Y)]}{\sigma_X \sigma_Y}$ as a measure of linear correlation between variable $X$ and $Y$, and we conduct this study on all predicted object pairs within each image in COCO `val` set averaged by all $5000$ images. The result is $\rho_{X,Y} = -0.80$. This means that $\boxed{\text{DET}}$ tokens that are close to each other (*i.e.*, with high cosine similarity) also lead to mostly nearby predictions (*i.e.*, with short $\ell_2$ distances, given $\rho_{X,Y} < 0$).

We also conduct a quantitative study on the relation between $X = $ the cosine similarity of $\boxed{\text{DET}}$ token pairs, and $Y = $ the corresponding cosine similarity of the output features of the classifier. The result is $\rho_{X,Y} = -0.07$, which is very close to $0$. This means that there is no strong linear correlation between these two variables.

**Detaching Detection Tokens.** To further understand the role $\boxed{\text{DET}}$ tokens plays, we study impacts caused by detaching the $\boxed{\text{DET}}$ tokens of YOLOS during training, *i.e.*, we don't optimize the parameters of the one hundred randomly initialized $\boxed{\text{DET}}$ tokens. As shown in Tab. 7, detaching the $\boxed{\text{DET}}$ tokens has a minor impact to AP. These results imply that $\boxed{\text{DET}}$ tokens mainly serve as the information carrier for the $\boxed{\text{PATCH}}$ tokens. Similar phenomena are also observed in Fang et al. [22].

| Model | $\boxed{\text{DET}}$ Tokens Config | AP |
|---|---|---|
| YOLOS-Ti | Rand. Init. & Learnable | 28.7 |
|  | Rand. Init. & **Detached** | 28.3 |
| YOLOS-S | Rand. Init. & Learnable | 36.1 |
|  | Rand. Init. & **Detached** | 36.4 |

Table 7: Impacts of detaching the $\boxed{\text{DET}}$ tokens of YOLOS during training.

## 4 Related Work

**Vision Transformer for Object Detection.** There has been a lot of interest in combining CNNs with forms of self-attention mechanisms [4] to improve object detection performance [9, 31, 63], while recent works trend towards augmenting Transformer with CNNs (or CNN design). Beal et al. [6] propose to use a pre-trained ViT as the feature extractor for a Faster R-CNN [50] object detector. Despite being effective, they fail to ablate the CNN architectures, region-wise pooling operations [23, 25, 27] as well as hand-crafted components such as dense anchors [50] and NMS. Inspired by modern CNN architecture, some works [39, 59, 62, 65] introduce the pyramidal feature hierarchy and locality to Vision Transformer design, which largely boost the performance in dense prediction tasks including object detection. However, these architectures are performance-oriented and cannot reflect the properties of the canonical or vanilla Vision Transformer [21] that directly inherited from Vaswani et al. [58]. Another series of work, the DEtection TRansformer (DETR) families [10, 72], use a random initialized Transformer to encode & decode CNN features for object detection, which does not reveal the transferability of a pre-trained Transformer.

UP-DETR [16] is probably the first to study the effects of unsupervised pre-training in the DETR framework, which proposes an "object detection oriented" unsupervised pre-training task tailored for Transformer encoder & decoder in DETR. In this paper, we argue for the characteristics of a pre-trained vanilla ViT in object detection, which is rare in the existing literature.

**Pre-training and Fine-tuning of Transformer.** The textbook-style usage of Transformer [58] follows a "pre-training & fine-tuning" paradigm. In NLP, Transformer-based models are often pre-trained on large corpora and then fine-tuned for different tasks at hand [18, 44]. In computer vision, Dosovitskiy et al. [21] apply Transformer to image recognition at scale using modern vision transfer learning recipe [33]. They show that a standard Transformer encoder architecture is able to attain excellent results on mid-sized or small image recognition benchmarks (*e.g*, ImageNet-$1k$ [51], CIFAR-10/100 [34], *etc.*) when pre-trained at sufficient scale (*e.g*, JFT-300M [55], ImageNet-$21k$ [17]). Touvron et al. [57] achieves competitive Top-1 accuracy by training Transformer on ImageNet-$1k$ only, and is also capable of transferring to smaller datasets [34, 42, 43]. However, existing transfer learning literature of Transformer arrest in image-level recognition and does not touch more complex tasks in vision such as object detection, which is also widely used to benchmark CNNs transferability.

Our work aims to bridge this gap. We study the performance and properties of ViT on the challenging COCO object detection benchmark [36] when pre-trained on the mid-sized ImageNet-$1k$ dataset [51] using different strategies.

## 5   Discussion

Over recent years, the landscape of computer vision has been drastically transformed by Transformer, especially for recognition tasks [10, 21, 39, 57, 59]. Inspired by modern CNN design, some recent works [39, 59, 62, 65] introduce the pyramidal feature hierarchy as well as locality to vanilla ViT [21], which largely boost the performance in dense recognition tasks including object detection.

We believe there is nothing wrong to make performance-oriented architectural designs for Transformer in vision, as choosing the right inductive biases and priors for target tasks is crucial for model design. However, we are more interested in designing and applying Transformer in vision following the spirit of NLP, *i.e.*, pre-train the *task-agnostic* vanilla Vision Transformer for general visual representation learning first, and then fine-tune or adapt the model on specific target downstream tasks *efficiently*. Current state-of-the-art language models pre-trained on massive amounts of corpora are able to perform few-shot or even zero-shot learning, adapting to new scenarios with few or no labeled data [8, 38, 45, 46]. Meanwhile, prevalent pre-trained computer vision models, including various Vision Transformer variants, still need a lot of supervision to transfer to downstream tasks.

We hope the introduction of Transformer can not only unify NLP and CV in terms of the architecture, but also in terms of the methodology. The proposed YOLOS is able to turn a pre-trained ViT into an object detector with the fewest possible *modifications*, but our ultimate goal is to adapt a pre-trained model to downstream vision tasks with the fewest possible *costs*. YOLOS still needs 150 epochs transfer learning to adapt a pre-trained ViT to perform object detection, and the detection results are far from saturating, indicating the pre-trained representation still has large room for improvement. We encourage the vision community to focus more on the general visual representation learning for the *task-agnostic* vanilla Transformer instead of the *task-oriented* architectural design of ViT. We hope one day, in computer vision, a universal pre-trained visual representation can be easily adapted to various understanding as well as generation tasks with the fewest possible *costs*.

## 6   Conclusion

In this paper, we have explored the transferability of the vanilla ViT pre-trained on mid-sized ImageNet-$1k$ dataset to the more challenging COCO object detection benchmark. We demonstrate that 2D object detection can be accomplished in a pure sequence-to-sequence manner with minimal additional inductive biases. The performance on COCO is promising, and these preliminary results are meaningful, suggesting the versatility and generality of Transformer to various downstream tasks.

## Acknowledgment

This work is in part supported by NSFC (No. 61876212, No. 61733007, and No. 61773176) and the Zhejiang Laboratory under Grant 2019NB0AB02. We thank Zhuowen Tu for valuable suggestions.

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
