# Appendix

**Position Embedding (PE) of YOLOS**

In object detection and many other computer vision benchmarks, the image resolutions as well as the aspect ratios are usually not fixed as the image classification task. Due to the changes in input resolutions & aspect ratios (sequence length) from the image classification task to the object detection task, the position embedding (PE) in ViT / YOLOS has also to be changed and adapted[1]. The changes in PE could affect the model size and performance. In this work, we study two types of PE settings for YOLOS:

- Type-I adds randomly initialized PE to the input of each intermediate Transformer layer as DETR [1], and the PE is 1D learnable (considering the inputs as a sequence of patches in the raster order) as ViT [3]. For the first layer, the PE is interpolated following ViT. The size of PEs is usually smaller than the input sequence size considering the model parameters. In the paper, small- and base-sized models use this setting.

- Type-II interpolates the pre-trained 1D learnable PE to a size similar to or slightly larger than the input size, and adds no PE in intermediate Transformer layers. In the paper, tiny-sized models use this setting.

In a word, Type-I uses more PEs and Type-II uses larger PE.

**Type-I PE.**    This setting adds PE to the input of each Transformer layer following DETR [1], and the PE considering the inputs as a sequence of patches in the raster order following ViT [3]. Specifically, during fine-tuning, the PE of the first layer is interpolated from the pre-trained one, and the PEs for the rest intermediate layers are randomly initialized and trained from scratch. In our paper, small- and base-sized models use this setting. The detailed configurations are given in Tab. 1.

| Model | PE-cls to PE-det @ First Layer | | Rand. Init. PE-det @ Mid. Layer | cls $\rightarrow$ det Params. (`M`) |
|---|---|---|---|---|
| YOLOS-S | $\frac{224}{16} \times \frac{224}{16}$ | $\nearrow \frac{512}{16} \times \frac{864}{16}$ | $\frac{512}{16} \times \frac{864}{16}$ | $22.1 \rightarrow 30.7$ |
| YOLOS-S ($dwr$) | $\frac{224}{16} \times \frac{224}{16}$ | $\nearrow \frac{800}{16} \times \frac{1344}{16}$ | $\frac{800}{16} \times \frac{1344}{16}$ | $13.7 \rightarrow 22.0$ |
| YOLOS-S ($d\mathbf{w}r$) | $\frac{224}{16} \times \frac{224}{16}$ | $\nearrow \frac{512}{16} \times \frac{864}{16}$ | $\frac{512}{16} \times \frac{864}{16}$ | $19.0 \rightarrow 27.6$ |
| YOLOS-B | $\frac{384}{16} \times \frac{384}{16}$ | $\nearrow \frac{800}{16} \times \frac{1344}{16}$ | $\frac{800}{16} \times \frac{1344}{16}$ | $86.4 \rightarrow 127.8$ |

Table 1: Type-I PE configurations for YOLOS models. "PE-cls $\nearrow$ PE-det" refers to performing 2D interpolation of ImageNet-$1k$ pre-trained PE-cls to PE-det for object detection. The PEs added in the intermediate (Mid.) layers (all the other layers of YOLOS except the first layer) are randomly initialized.

From Tab. 1, we conclude that it is expensive in terms of model size to use intermediate PEs for object detection. In other words, about $\frac{1}{3}$ of the model weights is for providing positional information only. Despite being heavy, we argue that the randomly initialized intermediate PEs do not directly inject additional inductive biases and they learn the positional relation from scratch. Nevertheless, for multi-scale inputs during training or input with different sizes & aspect ratios during inference, we (have to) adjust the PE size via 2D interpolation on the fly [2]. As mentioned in Dosovitskiy et al. [3] and in the paper, this operation could introduce inductive biases.

To control the model size, these intermediate PE sizes are usually set to be smaller than the input sequence length, *e.g.*, for typical models YOLOS-S and YOLOS-S ($d\mathbf{w}r$), the PE size is $\frac{512}{16} \times \frac{864}{16}$. Since the $dwr$ scaling is more parameter friendly compared with other model scaling approaches, we

---

[1]PE for one hundred `[DET]` tokens is not affected.

[2]There are some kind of data augmentations that can avoid PE interpolation, *e.g.*, large scale jittering used in Tan et al. [5], which randomly resizes images between $0.1\times$ and $2.0\times$ of the original size then crops to a fixed resolution. However, scale jittering augmentation usually requires longer training schedules, in part because when the original input image is resized to a higher resolution, the cropped image usually has a smaller number of objects than the original, which could weaken the supervision signal therefore needs longer training to compensate. So there is no free lunch.

use a larger PE for YOLOS-S ($dwr$) than other small-sized models to compensate for the number of parameters. For larger models such as YOLOS-Base, we do not consider the model size so we also choose to use larger PE.

Using 2D PE can save a lot of parameters, *e.g.*, DETR uses two long enough PE ($\text{Length} = 50$ for regular models and $\text{Length} = 100$ for DC5 models) for both $x$ and $y$ axes. We don't consider 2D PE in this work.

| Model | PE Type | PE-cls to PE-det @ First Layer | Rand. Init. PE-det @ Rest Layer | Params. (M) cls → det | AP |
|---|---|---|---|---|---|
| YOLOS-Ti | Type-I | $\frac{224}{16} \times \frac{224}{16} \nearrow \frac{512}{16} \times \frac{864}{16}$ | $\frac{512}{16} \times \frac{864}{16}$ | $5.7 \to 9.9$ | 28.3 |
|  | Type-II | $\frac{224}{16} \times \frac{224}{16} \nearrow \frac{800}{16} \times \frac{1344}{16}$ | No PE | $5.7 \to 6.5$ | 28.7 |
| YOLOS-S | Type-I | $\frac{224}{16} \times \frac{224}{16} \nearrow \frac{512}{16} \times \frac{864}{16}$ | $\frac{512}{16} \times \frac{864}{16}$ | $22.1 \to 30.7$ | 36.1 |
|  | Type-II | $\frac{224}{16} \times \frac{224}{16} \nearrow \frac{960}{16} \times \frac{1600}{16}$ | No PE | $22.1 \to 24.6$ | 36.6 |

Table 2: Some instantiations of Type-II PE. They are lighter and better than Type-I counterparts.

**Type-II PE.** Later, we find that interpolating the pre-trained PE at the first layer to a size similar to or larger than the input sequence length as the only PE can provide enough positional information, and is more efficient than using more smaller-sized PEs in the intermediate layers. In other words, it is redundant to use intermediate PEs given one large enough PE in the first layer. Some instantiations are shown in Tab. 2. In the paper, tiny-sized models use this setting. This type of PE is more promising, and we will make a profound study about this setting in the future.

**Self-attention Maps of YOLOS**

We inspect the self-attention of the $[\text{DET}]$ tokens that related to the predictions on the heads of the last layer of YOLOS-S. The visualization pipeline follows Caron et al. [2]. The visualization results are shown in Fig. 1 & Fig. 2. We conclude that:

- For a given YOLOS model, different self-attention heads focus on different patterns & different locations. Some visualizations are interpretable while others are not.

- We study the attention map differences of two YOLOS models, *i.e.*, the 200 epochs ImageNet-1$k$ [4] pre-trained YOLOS-S and the 300 epochs ImageNet-1$k$ pre-trained YOLOS-S. Note that the AP of these two models is the same (AP= 36.1). From the visualization, we conclude that for a given predicted object, the corresponding $[\text{DET}]$ token as well as the attention map patterns are usually different for different models.

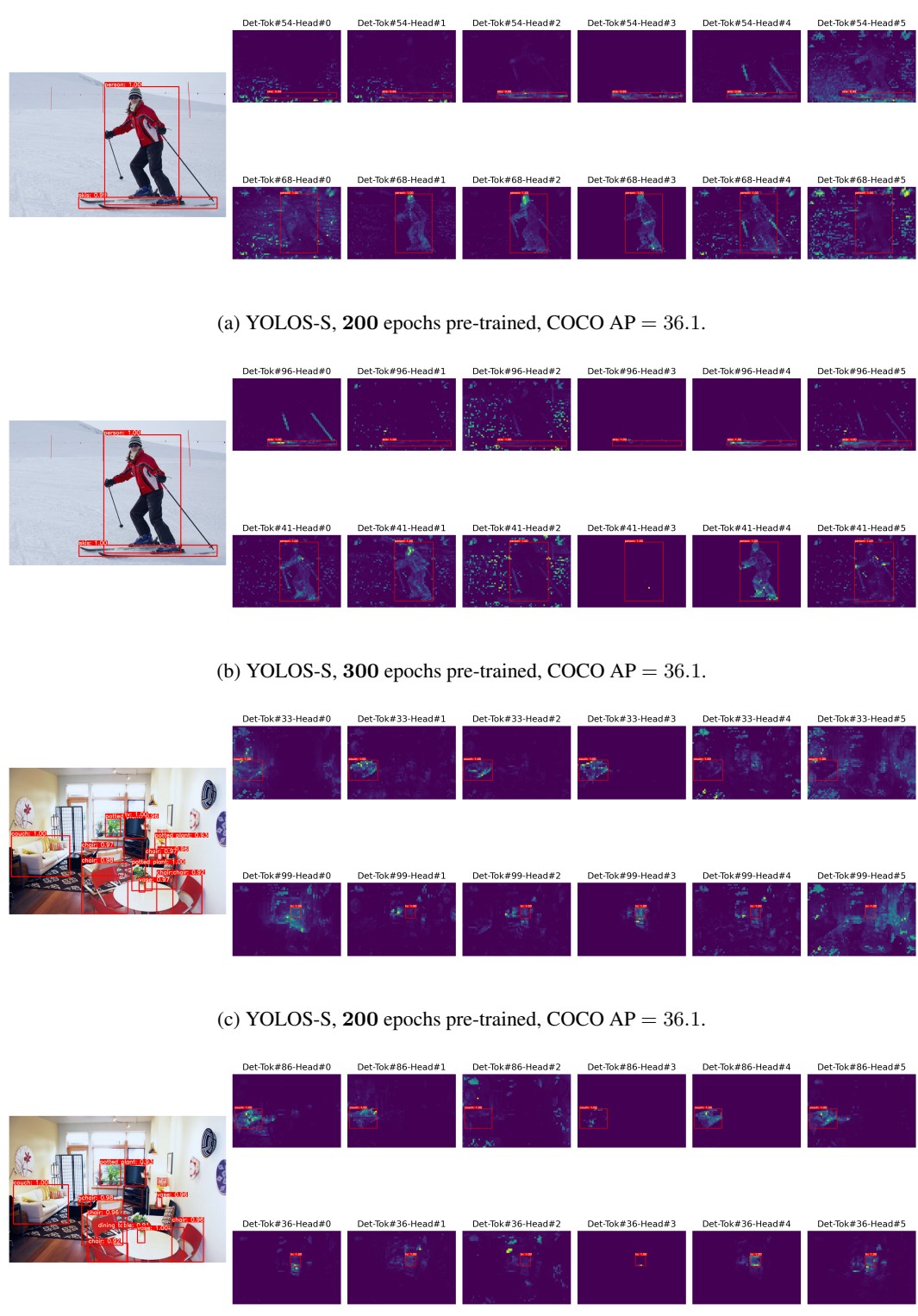

(a) YOLOS-S, **200** epochs pre-trained, COCO AP = 36.1.

(b) YOLOS-S, **300** epochs pre-trained, COCO AP = 36.1.

(c) YOLOS-S, **200** epochs pre-trained, COCO AP = 36.1.

(d) YOLOS-S, **300** epochs pre-trained, COCO AP = 36.1.

Figure 1: The self-attention map visualization of the [DET] tokens and the corresponding predictions on the heads of the last layer of two different YOLOS-S models.

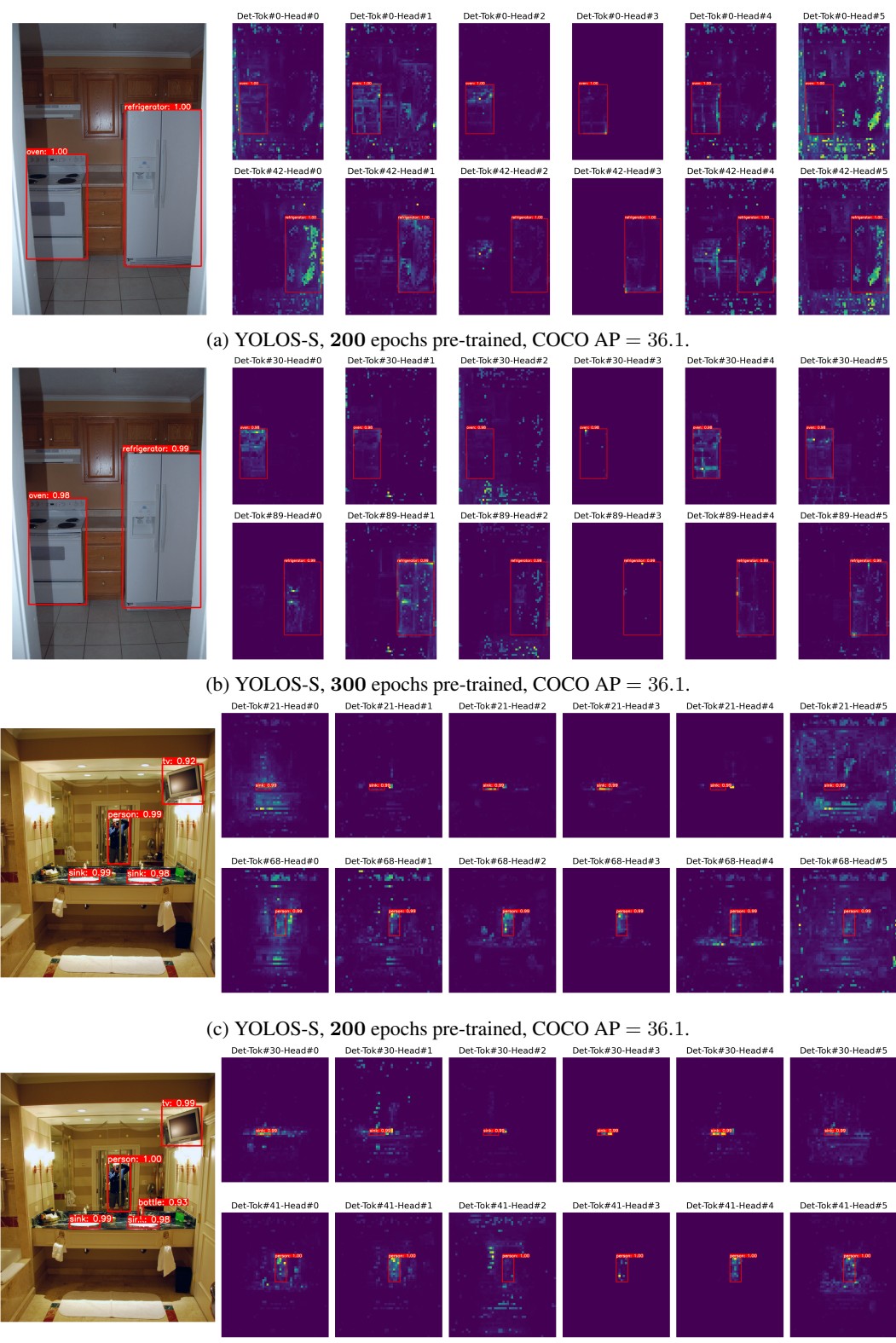

(a) YOLOS-S, **200** epochs pre-trained, COCO AP = 36.1.

(b) YOLOS-S, **300** epochs pre-trained, COCO AP = 36.1.

(c) YOLOS-S, **200** epochs pre-trained, COCO AP = 36.1.

(d) YOLOS-S, **300** epochs pre-trained, COCO AP = 36.1.

Figure 2: The self-attention map visualization of the [DET] tokens and the corresponding predictions on the heads of the last layer of two different YOLOS-S models.