# OpenReview forum: "You Only Look at One Sequence: Rethinking Transformer in Vision through Object Detection"
_NeurIPS.cc/2021/Conference — NeurIPS 2021 Poster_

### Official Review · Reviewer_hofm · 2021-07-14

**Rating:** 6
**Confidence:** 4

**Summary:**

This paper explores how to transfer the pure ViT that is pre-trained on ImageNet to the more challenging object detection task. To cope with the object detection task, the proposed YOLOS first drops the CLS tokens in ViT and appends learnable DET tokens. Besides, the bipartite matching loss is utilized to perform set prediction for objects. Extensive experiments are provided to analyze the impacts of pre-training schemes. The proposed YOLOS also shows good performance for object detection on COCO benchmark.

**Limitations And Societal Impact:**

The limitations have been discussed in the last section of the paper.

**Main Review:**

**Strengths**
-	The topic of this paper is actually quite simple yet interesting: could the vanilla pre-trained ViT be transferred to more complex task like object detection? Prior transformer-based object detection either utilizes transformer as feature extraction (ViT-FRCNN) or prediction head (DETR), whether transformer could perform object detection from sequence-to-sequence perspective is less explored and is thoroughly discussed in this paper.
-	Extensive experiments are provided to show how the pre-training schemes for ViT affect the performance of object detection.
-	The proposed YOLOS shows good performance on COCO.
**Weaknesses**
-	As discussed in Line 108, one of the main difference between the proposed YOLOS and DETR is that DETR uses a randomly initialized transformer while YOLOS utilizes pre-trained ViT. However, there are some prior works that discuss the pre-training of DETR, such as UP-DETR [Dai et. al, CVPR 2021]. What is the difference between the proposed method and these DETR pre-training method? Why the proposed method might be better?
-	There are also some prior methods that discuss and conclude that the decoder is not necessary for transformer-based object detection [Rethinking Transformer-based Set Prediction for Object Detection, arXiv 2020]. What is the different between the proposed YOLOS and these prior encoder-only transformer-based detectors?
-	Though the authors claim that YOLOS is not designed to be yet another high-performance  object detector, the performance is not quite competitive with state-of-the-arts detectors, especially compared with DETR.
-	I do not quite understand how the visualization in Figure 2 comes to the conclusion of “We observe that each [DET] token learns to specialize on certain regions and sizes.”
-	I am quite confused with the model scaling part. What does f(Lin.)/f(Att.) actually impact the scaling strategy? Can I conclude from the section “Towards Better Bigger Transformer” that the proposed scaling method is not suitable for YOLOS?

I have mixed evaluation for this manuscript and my initial rating is around borderline. On one hand, I do think that the topic discussed in this paper is quite interesting. But on the other hand, the proposed YOLOS does not present much novel design and the performance is also not quite strong.


**Time Spent Reviewing:**

4

---

> ### Author Response · Authors · 2021-08-10
> **Thanks and Response to Reviewer hofm (Part 2)**
>
> ***Question #4***: The visualization in Figure 2.
>
> ***Answer #4***: We give a more detailed description here: Figure 2 is the visualization of all box predictions on all images from COCO val split for the first $10$ [$\mathtt{DET}$] tokens. Each subfigure in Figure 2 corresponds to all bounding box predictions' centers of one [$\mathtt{DET}$] token in COCO val set. The points are color-coded: blue points correspond to small objects, green points correspond to medium objects, red points correspond to large objects. The object size definition follows the COCO dataset.
>
> In general, different subfigures in Figure 2 have different spatial modes and color preferences, which indicates that each [$\mathtt{DET}$​​​​] token learns to specialize on certain regions and sizes of objects.
>
> As a concrete example, from Figure 2 we observe #3 & #4 [$\mathtt{DET}$] tokens prefer to perceive **large** objects located near the **center**, while #6 [$\mathtt{DET}$] token trends to predict **small** objects located on the **left**.
>
>
> We will improve the description of this part and add some concrete examples in the revision.
>
> ------
>
> ***Question #5***:  What does $\frac{f(\mathtt{Lin}.)}{f(\mathtt{Att.})}$​​​ actually impact the scaling strategy?
>
> ***Answer #5***: In a word, $\frac{f(\mathtt{Lin}.)}{f(\mathtt{Att.})} \propto \frac{w}{r^2}$ is a measure of whether available model scaling strategies proved to be effective in CNNs [1, 2] are applicable to ViT / YOLOS: A CNN model scaling law is more applicable to ViT / YOLOS with higher $\frac{f(\mathtt{Lin}.)}{f(\mathtt{Att.})}$​ rate.
>
> The model complexity or FLOPs $(f)$​ of CNNs is $f(\mathtt{CNN}) \propto d w^2 r^2$​. Here $w$​ is the model width, $d$​ is the model depth and $r$​ is the input resolution following the notations in [2].
>
> Different from CNNs, the model complexity or FLOPs $(f)$​ of ViT / YOLOS is **a linear combination of two kinds of operations**. The first one is the linear projection ($\mathtt{Lin.}$​) or point-wise convolution, which fuses the information across different channels point-wisely via learnable parameters. The complexity is $f(\mathtt{Lin.}) \propto d w^2 r^2$​, which is the **same** as $f(\mathtt{CNN})$​. The second one is the spatial attention ($\mathtt{Att.}$​) part, which aggregates the spatial information depth-wisely via computed attention weights. The complexity is $f(\mathtt{Att.}) \propto d w r^4$​, which is **different** from $f(\mathtt{CNN})$​ and grows quadratically with the input sequence length or number of pixels.
>
> During the pre-training phase (i.e., ImageNet-$1k$​ image classification), the input resolution $r$​ is **relatively low**, therefore $f(\mathtt{Lin.})$​ **dominates** the FLOPs ($\frac{f\mathtt{(Lin.)}}{f(\mathtt{Att.})} > 5$​). Our experiments (in Table 3 of the manuscript) indicate that the model scaling properties of ViT (i.e., accuracy, FPS, FLOPs) are **consistent** with CNNs when $\frac{f\mathtt{(Lin.)}}{f(\mathtt{Att.})}$​ is large. Therefore previous CNNs model scaling strategies is still applicable during the pre-training phase  (i.e., ImageNet-$1k$​ image classification).
>
> During the finetuning or transfer learning phase (i.e., COCO object detection), the picture changes. The input resolution $r$​ is **much higher** (e.g., from $\sim 224$​ to $\sim 800$​) so the spatial attention part $f(\mathtt{Att.}) \propto d w r^4$​ takes over and linear projection part $f(\mathtt{Lin.}) \propto d w^2 r^2$​ is **no longer dominant** in terms of FLOPs. Canonical CNN model scaling recipes (e.g., [1, 2]) do not take spatial attention computations into account, so there **might be some inconsistencies** between pre-training and transfer learning performance, as shown in Table 3 and discussed in Sec 3.3 of our manuscript. Therefore during finetuning or transfer learning phase (i.e., COCO object detection), previous CNNs model scaling strategies are not directly applicable in principle, but empirically we find the $d \mathbf{w} r$​ scaling is still effective.
>
> [1] Tan et al., "EfficientNet: Rethinking Model Scaling for Convolutional Neural Networks". In ICML 2019.
>
> [2] Dollár et al., "Fast and Accurate Model Scaling". In CVPR 2021.
>
> ------
>
> ***Question #6***: What can we conclude from Section “Towards Better Bigger Transformer”?
>
> ***Answer #6***: We believe it is more like a Section about "future work". As mentioned in ***Answer #5***, Transformer blocks are more complicated than convolution kernels for there are two kinds of computations with different complexities, while there lacks systematic study on ViT model scaling law, especially under high input resolution. We hope this manuscript can serve as a catalyst for further study on ViT model scaling, and encourage more fellow researchers to investigate this interesting topic.
>
> ------
>
> ***Question #7***: YOLOS does not present much novel design.
>
> ***Answer #7***: We believe this is a matter of perspective. The motivation of this manuscript is to precisely & unbiasedly reveal the transferability of vanilla ViT in the object detection task via the simplest possible design, i.e., by only looking at one sequence (YOLOS). A sophisticated design with more inductive biases and prior knowledge of the task injected could make the experimental results be affected by both the original ViT and the newly added components **in a tangle**, which goes against our motivation.
>
> As agreed with **Reviewer R8B6**, YOLOS gets rid of the complex and diverse design of the detector head and presents a solution as neat as doing image classification to enable any available vanilla ViT performing object detection. Along with **Reviewer R8B6**, we also believe YOLOS takes a step toward unifying image classification and object detection tasks using Transformer. As you know, there is a growing trend towards unifying various tasks and modalities using a universal architecture with simple & compact task layers [1, 2], rather than using task-specific models, and Transformer is considered as a prime candidate. We hope the simple and effective design of YOLOS can contribute to this cause. Moreover, the implantation of YOLOS is very simple and does not require a specialized toolbox for object detection. We will release the code and models to facilitate future research.
>
> [1] Lu et al., "Pretrained Transformers as Universal Computation Engines". Arxiv 2103.05247.
>
> [2] Kim et al., "ViLT: Vision-and-Language Transformer Without Convolution or Region Supervision". In ICML 2021.

---

> > ### Comment · Reviewer_hofm · 2021-08-18
> > **Thanks for the response.**
> >
> > Thanks for the response and the rebuttal has addressed some of my concerns. I have one further question related to **Answer #3**, in which the authors show that scaling data and model size can bring consistent improvements for the proposed YOLOS. However, it is quite likely that other transfromer-based object detectors, e.g. DETR, may also beneficial from larger model and more data. From this point of view, the performance of YOLOS may be still not quite strong compared with other tranfromer-based detectors. I tend to keep my initianl rating.

---

> > > ### Author Response · Authors · 2021-08-19
> > > **Thanks for your feedback**
> > >
> > > Thanks for your feedback and glad to see some of your concerns have been addressed.
> > >
> > > ------
> > >
> > > Before answering your question, we would like to highlight our motivation and contribution here again: The motivation of designing YOLOS is to reveal the transfer learning properties of vanilla Transformer in object detection from a pure seq2seq perspective with minimal knowledge about the 2D spatial structure. To this end, YOLOS is not purposefully designed for better performance, but to precisely & unbiasedly reveal the real transferability of vanilla Transformer in object detection. Specifically:
> > >
> > > - We directly adopt canonical architecture widely used in NLP, e.g., BERT-Base and BERT-Large [1], which is **not** originally designed to tackle computer vision tasks like object detection.
> > > - We don't impose additional inductive biases and prior knowledge about the target task into the architecture, because adding heuristics could make the experimental results be affected by both the original ViT and the newly added components **in a tangle**, which goes against our motivation. So we choose to keep the whole framework simple.
> > >
> > > Therefore, the results of YOLOS can **precisely & unbiasedly** prove that the Transformer architecture, which has already shown impressive performance on a wide range of NLP tasks, can also directly adapt to complex computer vision tasks such as object detection. Moreover, YOLOS can be used as **a touchstone to benchmark** the transferability from image classification to object detection of different (label-supervised or self-supervised) pre-training strategies and architectural design of ViT. Despite not being SOTA, the results are very **meaningful**, because together with previous efforts (e.g., BERT [1], ViT [2], etc.) they exactly reveal the **versatility and generality** of vanilla Transformer, in both language and vision tasks.
> > >
> > > Note that the aforementioned conclusion cannot be drawn from existing Transformer-based object detectors, e.g., DETR [3], because they are usually CNN-Transformer hybrid architectures and they exploit additional 2D inductive biases about the target task. Therefore the performance of these hybrid architectures cannot directly reflect the true nature of vanilla Transformer.
> > >
> > > ------
> > >
> > > As for your concern about the model scaling effects in CNN-Transformer hybrid architecture, as pointed out in [our response to Reviewer R8B6](https://openreview.net/forum?id=nVofoXjTmA_&noteId=VohzYdw8qEH) and also in [2], under the low data regime and limited model size, the CNN-Transformer hybrid architecture largely benefit from the 2D inductive biases. However, with growing models and datasets sizes, the effects of these image-specific inductive biases **gradually vanish** [2]. Moreover, [2] also quantitatively shows that vanilla ViT is even **more computationally efficient** than the state-of-the-art BiT CNNs [4] as well as CNN-ViT hybrid architectures [2]. [5] also demonstrates that pre-trained CLIP-ViTs are better than CLIP-ResNets and other CNN models in large scale visual transfer learning. Therefore, we believe our YOLOS can also **benefit more** from larger models and more data than CNN-Transformer hybrid architectures.
> > >
> > > *As requested, we are now working on a preliminary experiment transferring a DETR model with ImageNet-21k pre-trained BiT ResNet [4] backbone to perform object detection. This large CNN-Transformer hybrid detector has a similar model size as YOLOS based on BERT-Large. This experiment takes about a weak. Please stay tuned.*
> > >
> > > ------
> > >
> > > At last, we would like to emphasize that there is a growing trend towards unifying various tasks and modalities using a universal architecture with simple & compact task layers [6, 7] and minimal inductive biases, rather than using separate task-specific models with a lot of task-specific inductive biases and heuristics. Since Transformer is considered as a prime candidate for this great cause, we hope the proposed YOLOS can serve as a catalyst and facilitate future research.
> > >
> > >
> > >
> > > References:
> > >
> > >
> > >
> > > [1] Devlin et al., "BERT: Pre-training of Deep Bidirectional Transformers for Language Understanding". In NAACL 2019.
> > >
> > > [2] Dosovitskiy et al., "An Image is Worth 16x16 Words: Transformers for Image Recognition at Scale". In ICLR 2021.
> > >
> > > [3] Carion et al., "End-to-End Object Detection with Transformers". In ECCV 2020.
> > >
> > > [4] Kolesnikov et al., "Big transfer (BiT): General visual representation learning". In ECCV 2020.
> > >
> > > [5] Radford et al., "Learning Transferable Visual Models From Natural Language Supervision". Arxiv 2103.00020.
> > >
> > > [6] Lu et al., "Pretrained Transformers as Universal Computation Engines". Arxiv 2103.05247.
> > >
> > > [7] Kim et al., "ViLT: Vision-and-Language Transformer Without Convolution or Region Supervision". In ICML 2021.

---

> > > > ### Author Response · Authors · 2021-08-26
> > > > **Response to the concern about the scaling effects**
> > > >
> > > > As requested, we conduct a preliminary experiment transferring a DETR [1] system with ImageNet-$21k$ pre-trained BiT ResNet [2] backbone to perform object detection. The results of DETR and YOLOS are shown as follows:
> > > >
> > > > | Method | Backbone       | Pre-train Data|     Params.     | Pre-train & Fine-tune Total exaFLOPs ($\times 10 ^ {18}$) | AP          |
> > > > | :----- | -------------- | --------------- | :--------------: | :-------------------------------------------------------: | :---------- |
> > > > | DETR [1]   | BiT ResNet-101x3 [2] | ImageNet-$21k$ | $406 \mathrm{M}$ |                     $1.7 \times 10^3$                     | $44.5$    |
> > > > | YOLOS  | BERT-Large [3] | ImageNet-$21k$ | $311\mathrm{M}$ |                     $1.6 \times 10^3$                     | $45.1 (+0.6)$ |
> > > >
> > > > The two models are controlled to consume similar total computations. Our preliminary trials indicate that Transformer can benefit more from larger models and more data compared with CNNs, which are consistent with the conclusions from [4] in image recognition.
> > > >
> > > > We will include these discussions in the revision and hope our response can address your concern.
> > > >
> > > > ------
> > > >
> > > > References:
> > > >
> > > >
> > > >
> > > > [1] Carion et al., "End-to-End Object Detection with Transformers". In ECCV 2020.
> > > >
> > > > [2] Kolesnikov et al., "Big transfer (BiT): General visual representation learning". In ECCV 2020.
> > > >
> > > > [3] Devlin et al., "BERT: Pre-training of Deep Bidirectional Transformers for Language Understanding". In NAACL 2019.
> > > >
> > > > [4] Dosovitskiy et al., "An Image is Worth 16x16 Words: Transformers for Image Recognition at Scale". In ICLR 2021.

---

> ### Author Response · Authors · 2021-08-10
> **Thanks and Response to Reviewer hofm (Part 1)**
>
> We would like to thank you for your detailed comments to help us improve YOLOS, and we will improve our manuscript correspondingly. The responses to the main concerns are as follows.
>
> ------
>
> ***Question #1***: The difference between YOLOS and UP-DETR [1].
>
> ***Answer #1***: YOLOS and UP-DETR are essentially two different types of works. In a word, UP-DETR belongs to the works that **improve** the object detection performance of DETR (DETR is already an object detector) via injecting proper inductive biases, while YOLOS **enables** ViT to perform object detection (ViT is not an object detector) with the minimal possible inductive biases.
>
> The specific analyses are as follows. UP-DETR proposes an "object detection friendly" unsupervised pre-training task tailored for Transformer encoders & decoders in the DETR framework. It is essentially a CNN-Transformer hybrid architecture that exploits a pre-trained CNN to extract visual representation for Transformer encoders as inputs, and a pre-trained CNN to extract patch features for Transformer decoders as inputs during the unsupervised pre-training processes. Therefore UP-DETR still relies on CNN along with its inductive biases. Moreover, the unsupervised pretext task named random query patch detection also imposes image-specific inductive biases to Transformer encoders & decoders. A recent follow-up work DETReg [2] takes a step further and shows that adding more object detection related inductive biases and region priors during unsupervised pre-training is more beneficial. Overall, UP-DETR and DETReg improve DETR performance via injecting proper inductive biases.
>
> The proposed YOLOS focuses on transferring any available pre-trained ViT to object detection task in a pure seq2seq manner with minimal knowledge about the 2D spatial structure. Among existing object detectors, YOLOS utilizes the minimal possible 2D inductive biases and makes the fewest possible modiﬁcations on ViT to our knowledge. Therefore many pre-trained ViT models (e.g., [3] releases more than $50000$​​​ pre-trained ViT models trained under diverse settings on various datasets) can be used almost out of the box and directly be transferred to perform object detection via YOLOS.
>
> We would like to emphasize that YOLOS is not a pre-training scheme, but a transfer learning pipeline designed for ViT. In fact, YOLOS can be used as a touchstone or probe to benchmark and evaluate the transferability from image classification to object detection of different (label-supervised or self-supervised) pre-training strategies and architectural design of ViT, since we find that the object detection performance of ViT is quite sensitive to different pre-train schemes, as stated in line 195-201 of the manuscript.
>
> We will add and discuss these works in our revised version for a more thorough literature review and to avoid misunderstandings.
>
> [1] Dai et al., "UP-DETR: Unsupervised Pre-training for Object Detection with Transformers". In CVPR 2021.
>
> [2] Bar et al., "DETReg: Unsupervised Pretraining with Region Priors for Object Detection". Arxiv 2106.04550.
>
> [3] Steiner et al., "How to train your ViT? Data, Augmentation, and Regularization in Vision Transformers". Arxiv 2106.10270.
>
> ------
>
> ***Question #2***: [1] concludes "decoder is not necessary for transformer-based object detection" & The difference between YOLOS and the encoder-only transformer-based detector [1].
>
> ***Answer #2***: First, we believe the conclusion "decoder is not necessary for transformer-based object detection " from [1] is somewhat controversial since DETR [2] actually has already discussed & studied the necessity of decoders and concluded:
>
> > "We analyze the importance of each decoder layer by evaluating the objects that would be predicted at each stage of the decoding (Fig. 4). Both AP and AP$_{50}$​ improve after every (decoder) layer, totalling into a very signiﬁcant +$8.2/9.5$​​ AP improvement between the ﬁrst and the last (decoder) layer. "
> >
> > ​															---- Excerpt from Page 11 in "End-to-End Object Detection with Transformers" [2]
>
> Therefore DETR concludes that decoders are crucial for a good performance. We hypothesize the controversial conclusions in [1] come from a $40$ epochs short schedule experiment (shown in Figure 4 of [1], which is too short for DETR to converge) therefore cannot generalize well. (To our knowledge, [1] hasn't been peer-reviewed and published so far)
>
> Second, YOLOS and TSP-FCOS & TSP-RCNN object detectors proposed in [1] are two different types of works in essence. As its name implies, TSP-FCOS & TSP-RCNN basically borrow the best practice of modern object detector design (e.g., FPN, FCOS, RCNN) and replace the dense prior heads with DETR Transformer encoder, and use set prediction loss to eliminate NMS. Many region priors (e.g., RPN, RoI) and heuristics (e.g., IoU threshold, $700$​ object candidates) are still involved. Moreover, some more recent peer-reviewed works [3, 4, 5] show that the Transformer encoder is not necessary for FCOS & RCNN to become end-to-end and achieve quite competitive performance.
>
> Overall, the aforementioned works [1, 3, 4, 5] focus on enabling modern object detectors (FCOS, RCNN) to become end-to-end using set prediction from DETR. Meanwhile, our YOLOS focuses on enabling vanilla Transformers, which are originally language models (e.g., BERT, GPT) or image classification models (e.g., ViT, DeiT), to perform the challenging object detection task with the minimal possible knowledge about the 2D spatial structure and the fewest possible modifications & inductive biases.
>
> We will add and discuss these works in our revised version for a more thorough literature review and to avoid misunderstandings.
>
> [1] Sun et al., "Rethinking Transformer-based Set Prediction for Object Detection". Arxiv 2011.10881.
>
> [2] Carion et al., "End-to-End Object Detection with Transformers". In ECCV 2020.
>
> [3] Sun et al., "What Makes for End-to-End Object Detection?". In ICML 2021.
>
> [4] Wang et al., "End-to-End Object Detection with Fully Convolutional Network". In CVPR 2021.
>
> [5] Sun et al., "Sparse R-CNN: End-to-End Object Detection with Learnable Proposals". In CVPR 2021.
>
> ------
>
> ***Question #3***: The performance of YOLOS is good but not quite strong.
>
> ***Answer #3***: As pointed in [1], Transformer / ViT usually benefits from more data and larger models. Restricted by the resources, in our manuscript we only investigate pre-training using mid-sized ImageNet-$1k$​ dataset, which already achieves good transfer learning performance (i.e., YOLOS-Base directly adopted from BERT-Base architecture achieves $42.0$​​ box AP on COCO). Here we add some preliminary trials on scaling data and model size.
>
> | Model                                    | Pre-train                               | Finetune COCO AP (150 Epochs) |
> | :--------------------------------------- | --------------------------------------- | :---------------------------: |
> | YOLOS (BERT-Base Like Architecture [2])  | ImageNet-$1k$​ (DeiT Like Pre-train [3]) |            $42.0$             |
> | YOLOS (BERT-Base Like Architecture [2])  | ImageNet-$21k$​ (ViT Like Pre-train [1]) |            $43.8$             |
> | YOLOS (BERT-Large Like Architecture [2]) | ImageNet-$21k$​​ (ViT Like Pre-train [1]) |            $45.1$             |
>
> The results suggest that scaling data and model size can bring consistent improvements and more competitive AP. We believe it is very promising to use even larger and stronger ViT / YOLOS models (e.g., [4, 5, 6]) to explore and push the boundary and take a step toward unifying various tasks using Transformer.
>
> [1] Dosovitskiy et al., "An Image is Worth 16x16 Words: Transformers for Image Recognition at Scale". In ICLR 2021.
>
> [2] Devlin et al., "BERT: Pre-training of Deep Bidirectional Transformers for Language Understanding". In NAACL 2019.
>
> [3] Touvron et al., "Training data-efficient image transformers & distillation through attention". In ICML 2021.
>
> [4] Steiner et al., "How to train your ViT? Data, Augmentation, and Regularization in Vision Transformers". Arxiv 2106.10270.
>
> [5] Zhai et al., "Scaling Vision Transformers". Arxiv 2106.04560.
>
> [6] Bao et al., "BEiT: BERT Pre-Training of Image Transformers". Arxiv 2106.08254.

---

### Official Review · Reviewer_R8B6 · 2021-07-15

**Rating:** 8
**Confidence:** 5

**Summary:**

This paper presents the design and evaluation of a Transformer-based object detector which works in a pure sequence-to-sequence manner. The proposed YOLOS detector is built from ViT by borrowing the [DET] token from detection transformer (DETR). A rather interesting result is that such a pure seq-to-seq architecture, which only takes non-overlapping image patches as input, can do object detection with a reasonable accuracy.

**Limitations And Societal Impact:**

This reviewer does not see any serious limitations or potential negative societal impact of this work.

**Main Review:**

Object detection is one of the fundamental problems in CV and image understanding. However, the solution to this problem is not as neat as image classification. It would be nice if an object detector can get rid of the complex and diverse design of the detector head. This paper presents a neat design of object detector and shows that a pre-trained Transformer learned from image classification task can be transferred to object detection to achieve a reasonable performance. This reviewer finds this paper quite interesting and is a meaningful investigation toward a unified Transformer for image classification and object detection tasks.

However, there is one point of view that this reviewer cannot agree with the authors. The authors mentioned that some recent works [34, 53, 55] introduce the pyramidal feature hierarchy and locality to Vision Transformer design, which largely boost the performance. But the authors think that these architectures are performance-oriented and cannot reflect the properties of the vanilla ViT. This reviewer thinks that there is nothing wrong to make performance-oriented design for vision transformers. Images have their own characteristics including locality and large scale span. It does not make much sense to stick to the vanilla ViT only because it is the first vision Transformer design or it has high similarity with the Transformer used in NLP. This reviewer thinks that the authors should encourage fellow researchers to investigate the proposed YOLOS on other more efficient vision Transformers.

== after rebuttal ==
The authors have addressed my concerns in the rebuttal, and they agree to include the related discussions in the final version. Besides, the authors provide convincing additional experimental results in the response to other reviewers. Taking all these into consideration and after the calibration with other papers I reviewed, I decided to raise my score.

**Time Spent Reviewing:**

5

---

> ### Author Response · Authors · 2021-08-10
> **Thanks and Response to Reviewer R8B6**
>
> We would like to thank you for your detailed comments to help us improve YOLOS, and we will improve our manuscript correspondingly. The responses to the main concerns are as follows.
>
> ------
>
> We fully agree with the Reviewer that there is nothing wrong to make performance-oriented designs for vision Transformers, as choosing the right inductive biases and priors for target tasks is crucial for model design. [1, 2, 3] show that under proper inductive biases (pyramidal feature hierarchy, locality, etc.) and strong regularizations, efficient vision Transformer variants with only $10^1 \mathrm{M} \sim 10^2 \mathrm{M}$ numbers of parameters can achieve very competitive performance and generalize quite well even with limited data.
>
> However, as [4] points out, the effectiveness of inductive biases depends on the model size and data regime for ViT. With growing models and datasets, the effects of image-specific inductive biases **gradually vanish** [4], and vanilla Transformer [5, 6] can achieve state-of-the-art performance under large model size and high data regime (e.g., ImageNet-$21k$​, JFT-$300 \mathrm{M}$​​, or knowledge from DALL-E encoder [7] used by [6]). Moreover, [4] points out that the vanilla ViT models need to be pre-trained only once, then their fine-tuning on a custom dataset of interest can be done with modest computational resources in a data-efficient manner. [4] also quantitatively shows that vanilla ViT is even more computationally efficient than BiT CNNs [8] as well as CNN-ViT hybrid architecture [4]. Therefore the resources needed for large vanilla ViT are still tractable.
>
> Overall, we believe it is more appropriate to encourage fellow researchers to investigate YOLOS on both vanilla Transformer and its efficient variants conditioning on the model size, data and application scenarios. We will add these discussions in the revision.
>
>
>
> [1] Vaswani et al., "Scaling Local Self-Attention for Parameter Efficient Visual Backbones". In CVPR 2021.
>
> [2] Liu et al., "Swin Transformer: Hierarchical Vision Transformer using Shifted Windows". In ICCV 2021.
>
> [3] Wang et al., "Pyramid Vision Transformer: A Versatile Backbone for Dense Prediction without Convolutions". In ICCV 2021.
>
> [4] Dosovitskiy et al., "An Image is Worth 16x16 Words: Transformers for Image Recognition at Scale". In ICLR 2021.
>
> [5] Zhai et al., "Scaling Vision Transformers". Arxiv 2106.04560.
>
> [6] Bao et al., "BEiT: BERT Pre-Training of Image Transformers". Arxiv 2106.08254.
>
> [7] Ramesh et al., "Zero-Shot Text-to-Image Generation". Arxiv 2102.12092.
>
> [8] Kolesnikov et al., "Big transfer (BiT): General visual representation learning". In ECCV 2020.

---

> > ### Comment · Reviewer_R8B6 · 2021-08-21
> > **Thanks for the response**
> >
> > This reviewer agrees with the points the authors made in the response. It would be good to include these discussions in the final version.

---

> > > ### Author Response · Authors · 2021-08-21
> > > **Thanks for the feedback**
> > >
> > > Thanks for your feedback and glad to see your concerns have been addressed.
> > >
> > > We will include these discussions in the final version.

---

### Official Review · Reviewer_yaEX · 2021-07-16

**Rating:** 7
**Confidence:** 4

**Summary:**

The paper suggests a simple way of turning a classification pre-trained ViT into a detection model, without imposing any more image-specific inductive biases than the original ViT model itself. This is achieved by appending 100 [det] tokens to the input, and then optimally matching these with the ground-truth in the loss during training. The results are competitive with, but do not exceed SOTA.

**Main Review:**

I think this is a good paper and do not have much criticism. The writing is upfront about not aiming for SOTA but seeing how far one can go with an as "objective" as possible architecture, and show that indeed one can go far!

1) The few sections about model scaling seem out of place. They do matter at all for the paper's main message. They reads as if it was something someone tried, is "kinda meh", but does not want to throw away the results. I would have preferred if they were not included, but more analysis/details of the model was

2) An interesting thing to analyze would be, what is learned for the [det] tokens. For example, do those which are close to eachother (in latent space) also lead to mostly nearby detections?

3) As the first sentence mentions: "transformers are born to transfer" ; it would have been interesting to see how transformers pre-trained on larger datasets, such as ImageNet-21k, perform.

4) It is interesting that with this scheme, a DINO pre-trained ViT, which in the DINO paper is shown to have much better "spatial alignment", does not really lead to better detection results. I think this is worth explicitly highlighting.

5) Nitpick: footnote 3 belongs to social media, not in a paper.

Overall, my comments are not really shortcomings of the method, but ways in which the paper could be changed to turn it from "ok" to "great", in my opinion.

**Time Spent Reviewing:**

3

---

> ### Author Response · Authors · 2021-08-10
> **Thanks and Response to Reviewer yaEX**
>
> We would like to thank you for your detailed comments to help us improve YOLOS, and we will improve our manuscript correspondingly. The responses to the main concerns are as follows.
>
> ------
>
> ***Question #1***: Issues with the model scaling sections.
>
> ***Answer #1***: Thanks for the kind suggestion. We agree that several model scaling sections seem a little wordy, and we will rephrase them to one section focusing more on the details & analysis of different scaled models.
>
> ------
>
> ***Question #2***: An interesting thing to analyze would be, what is learned for the [$\mathtt{DET}$​​] tokens. For example, do those which are close to each other (in latent space) also lead to mostly nearby detections?
>
> ***Answer #2***: Thanks for your valuable question. We use the **Pearson correlation coefficient** $\rho_{X, Y}=\frac{\mathbb{E}\left[\left(X-\mu_{X}\right)\left(Y-\mu_{Y}\right)\right]}{\sigma_{X} \sigma_{Y}}$​ as a measure of linear correlation between variable $X$​ and $Y$​. The correlation coefficient $\rho_{X, Y}$​ ranges from $−1$​ to $+1$​.  An absolute value of exactly $1$​ implies that a linear equation perfectly describes the relationship between $X$​ and $Y$​, with all data points lying on a line. $\rho_{X, Y} = +1$​ implies that all data points lie on a line for which $Y$​ increases as $X$​ increases, and $\rho_{X, Y} = -1$​ implies that all data points lie on a line for which $Y$​ decreases as $X$​​ increases.
>
> Here we give a quantitative analysis on the relation between $X = $​​  the cosine similarity of [$\mathtt{DET}$​​] token pairs, and $Y = $​​  the corresponding predicted bounding box centers euclidean distances.  We conduct this study on all predicted object pairs within each image in COCO val set averaged by all $5000$​​ images. The result is $\rho_{X, Y} = -0.7995$​​. This means that [$\mathtt{DET}$​​] tokens that are close to each other (i.e., with **high** cosine similarity) also lead to mostly nearby detections (i.e., with **short** euclidean distances, given $\rho_{X, Y} < 0$​​​​). This result verifies your hypothesis.
>
> We also conduct a quantitative study on the relation between $X = $​  the cosine similarity of [$\mathtt{DET}$​] token pairs, and $Y = $​  the corresponding cosine similarity of the output features of the classifier using the Pearson correlation coefficient $\rho_{X, Y}$​. The result is $\rho_{X, Y} = -0.0693$​, which is very close to $0$. This means that there is no strong linear correlation between these two variables.
>
> We will add these analyses in the revision.
>
> ------
>
> ***Question #3***: It would have been interesting to see how transformers pre-trained on larger datasets, such as ImageNet-$21k$​, perform.
>
> ***Answer #3***: Here we add some preliminary trials on scaling data and model size.
>
> | Model                                    | Pre-train                               | Finetune COCO AP (150 Epochs) |
> | :--------------------------------------- | --------------------------------------- | :---------------------------: |
> | YOLOS (BERT-Base Like Architecture [1])  | ImageNet-$1k$ (DeiT Like Pre-train [2]) |            $42.0$             |
> | YOLOS (BERT-Base Like Architecture [1])  | ImageNet-$21k$​ (ViT Like Pre-train [3]) |            $43.8$             |
> | YOLOS (BERT-Large Like Architecture [1]) | ImageNet-$21k$​ (ViT Like Pre-train [3]) |            $45.1$             |
>
>
> The results suggest that scaling data and model size can bring consistent improvements and more competitive AP. We believe it is very promising to use even larger and stronger ViT / YOLOS models (e.g., [4, 5, 6]) to explore and push the boundary and take a step toward unifying various tasks using Transformer.
>
> [1] Devlin et al., "BERT: Pre-training of Deep Bidirectional Transformers for Language Understanding". In NAACL 2019.
>
> [2] Touvron et al., "Training data-efficient image transformers & distillation through attention". In ICML 2021.
>
> [3] Dosovitskiy et al., "An Image is Worth 16x16 Words: Transformers for Image Recognition at Scale". In ICLR 2021.
>
> [4] Steiner et al., "How to train your ViT? Data, Augmentation, and Regularization in Vision Transformers". Arxiv 2106.10270.
>
> [5] Zhai et al., "Scaling Vision Transformers". Arxiv 2106.04560.
>
> [6] Bao et al., "BEiT: BERT Pre-Training of Image Transformers". Arxiv 2106.08254.
>
> ------
>
> ***Question #4***: It is interesting that with this scheme, a DINO pre-trained ViT, which in the DINO paper is shown to have much better "spatial alignment", does not really lead to better detection results. I think this is worth explicitly highlighting.
>
> ***Answer #4***: Thanks for the suggestion and we will highlight this part in the revision.
>
> ------
>
> ***Question #5***: footnote 3 belongs to social media, not in a paper.
>
> ***Answer #5***: Thanks and we will improve the footnotes in the revision.

---

> > ### Comment · Reviewer_yaEX · 2021-08-29
> > **Thank you**
> >
> > Thank you for your answers. Indeed, the ImageNet-21k results look quite promising! I recommend adding them to the paper.

---

> > > ### Author Response · Authors · 2021-08-30
> > > **Thanks for the feedback**
> > >
> > > Thanks for your feedback and we will improve our manuscript accordingly.

---

### Decision · Program_Chairs · 2021-09-27

**Decision:**

Accept (Poster)

**Comment:**

The authors present a simple algorithm for converting a Vision Transformer (ViT) trained for classification into a detection model. The approach is based on dropping the the CLS tokens in ViT and appends learnable DET tokens, followed by matching these to the ground-truth during training. Empirical results are not SOTA, but are competitive. The reviewers appreciate the simplicity and the relatively strong performance of the resulting model. During the discussion the reviewers agreed that the rebuttal clarified the remaining points. I will hence recommend acceptance. I urge the authors to include suggested improvements in the revised version.